
# A plasmachemical axially symmetric self-consistent model of daytime sprite

Andrey Evtushenko[1], Fedor Kuterin[1], and Ekaterina Svechnikova[1]

[1]Federal Research Center Institute of Applied Physics of the Russian Academy of Sciences (IAP RAS), 46 Ul'yanov Street, Nizhny Novgorod, 603950, Russia

**Correspondence:** A. Evtushenko (a_evtushenko@inbox.ru)

**Abstract.** The paper presents the results of self-consistent axially symmetric modelling of a daytime sprite. Perturbations of the concentrations of ions, neutral compounds, excited atoms and molecules along with disturbances of the atmospheric conductivity and electric field due to the initiation of a daytime sprite are studied. It is shown that in daytime conditions a sprite develops in the altitude range between 50 and 70 km, which is approximately 20 km lower than the range of a typical

nighttime sprite. The uncompensated charge of a parent flash is typically characterized by the impulse charge moment (ICM) of several thousands of C·km; in the modelling of a daytime sprite ICM values are assumed to lie in the range of 2000–4000 C·km. It is shown that the behaviour of the system can be described by two scenarios, with and without a rapid increase in electron concentration, which are studied in detail by assuming ICM values of 3750 C·km and 2750 C·km respectively. During a discharge with an ICM of 2750 C·km, the decrease in the concentration of electrons in the electric field is caused by

their attachment to molecular oxygen, and no sharp increase in electron concentration occurs; the concentrations of the most significant ions and electrons reach unperturbed values in less than a second. For an ICM of 3750 C·km, an initial decrease in the electron concentration is followed by the formation of an avalanche of electrons characterized by an increase in their concentration by more than an order of magnitude relative to the initial value. A slight decrease in the electric field leads to another sharp decrease in the electron concentration; then relaxation of the ion concentration makes the electron concentration

increase, but it is not before about one second after the discharge that the latter once again reaches the maximum value attained during the avalanche, and then it does not change much for several tens of second. A rapid increase in electron concentration occurs in the central part of a sprite, leading to the conductivity disturbance and rapid displacement of the electric field. As a result of this study, the possibility of the initiation of a daytime sprite by an extremely intense lightning discharge leading to a significant long-term perturbation of atmospheric chemical balance is demonstrated.

# 1 Introduction

An active study of high-altitude discharges has been going on for more than 25 years, but many questions of the initiation of discharges and their effect on the chemical composition of the atmosphere remain unresolved. Sprites are of especial interest because of high frequency of their appearance (at least 1 event per minute (Chen A.B., 2008)), large volume up to $10^4 km^3$ (Pasko V.P. (2010), Liu N. (2015)) and the significant local influence on the conductivity of the mesosphere and lower





ionosphere. The sprite initiation is associated with the formation of an uncompensated charge in the cloud after a powerful cloud-to-ground discharge, usually of positive polarity. Under the assumption of the high conductivity of the Earth's surface the electric field of the uncompensated charge could be described in the dipole approach and decreases with the increase of the distance according to the power law. For a sprite development the electric field in the troposphere is required to exceed the breakdown field which decreases exponentially with increasing altitude above the Earth's surface, like atmospheric air 30 pressure. Therefore, the main parameters determining the possibility of initiation of a sprite is the magnitude of the transferred charge and the height of transfer.

A lot of observational data on nighttime sprites has been accumulated (Lu G., 2017). Characteristic values of the ICM for initiation a nighttime sprite are 400–600 C·km, while the minimum possible ICM capable of initiating a sprite is about 120 C·km, and the maximum recorded values reach 1000 C·km or more (Hu W., 2002). Theoretical estimates show that the 35 minimum ICM is about 200 C·km for positive cloud-to-ground flashes, and 320 C·km for negative, and the presence of an initial perturbation of conductivity is important (Qin J., 2013). The initiation of sprites is probabilistic, and the high ICM value is only a necessary condition. Nevertheless, the ICM of 500 C·km at night with almost 100% probability leads to the initiation of a high-altitude discharge from a positive cloud-to-ground discharge (Lu G., 2013).

The possibility of a sprite initiation in daytime conditions remains unclear, primarily due to the fact that it is impossible to 40 detect an optical flash, because the brightness of the flash is too low compared to the background glow of the sky. The existence of daytime sprites is indicated by indirect electromagnetic measurements. According to measurement results, the magnetic field dynamics typical of nighttime sprites are characterised by two peaks: the first peak of the field is related to the parent lightning flash and the second peak corresponds to a sprite (Cummer S.A., 2003). Similar pattern of the magnetic field evolution was observed in daytime and is presumably related to sprites (Kumar S., 2008). The estimated ICM is 2000–5000 C·km for a 45 daytime sprite, the corresponding charge transferred by the parent lightning discharge exceeds the value typical of nighttime sprite by 5–10 times. The difference is associated with a change in the possible altitude of initiation of a high-altitude discharge. In the daytime, the ionosphere conductivity at altitudes of 70–90 km is higher than at night by about 2 orders of magnitude, which complicates the penetration of the electric field into this region. A decrease in conductivity is observed below 70 km and the initiation of sprites occurs lower — in areas with a higher atmospheric density and a higher breakdown field value, which 50 means that the initiation of a sprite during the day requires a much greater ICM than at night (Fernsler R.F. (1996), Stanley M. (2000)). A daytime sprite altitude, compared with night conditions, shifts by 20 km downward, and consequently, the pressure in the sprite region increases by about an order of magnitude. Thus, for the daytime sprite initiation the required ICM by an order of magnitude higher than at nighttime, because the same value of the electric field normalized to concentration of neutral compounds is needed. Consequently, the diffuse part of the daytime sprite should be simulated in the altitude range of 50–70 55 km with the ICM values in the range 2000–5000 C·km.



## 2 Model of Initiation and Development of a Sprite

The numerical modeling of the influence of high-altitude discharges on the chemical composition of the atmosphere has been carried out for various conditions (Mishin E.V. (2008), Sentman D.D. (2008), Gordillo-Vázquez F.G. (2008), Evtushenko A.A. (2017), Lu G. (2013), Evtushenko A.A. (2013), Kotovsky D.A. (2016)). In all cases, the formulation of the simulation approach

begins with determining the list of species and setting initial conditions for it in the entire range of altitudes. At the next stage, for all species it is necessary to choose the reactions defining the concentration dynamics (sources and sinks). Model and experimental data allow us to find out only concentrations of the main neutral species containing nitrogen and oxygen, electrons and several ions, while experimental data are usually available only for certain altitudes and conditions (CESM, Mitra A.P. (1974)). Most complete data on the processes occurring in the nitrogen-oxygen mixture under the influence of an

electric field are available in (Kossyi I., 1992) and are used in all the models mentioned above. Unfortunately, (Kossyi I., 1992) does not contain data on reaction rates for chemical species containing carbon, chlorine, etc., which may be important at altitudes of interest. Thus, a set of chemical species used in modeling is limited by the accessibility of data on its concentration and chemical reactions.

Authors previously developed a self-consistent plasmachemical axially symmetric model of a nighttime sprite (Evtushenko A.A.,

2017). The model was adapted for the daytime sprites: a block of photochemical reactions is added, several ions and corresponding reactions are left beyond the framework of consideration, new values for the initial concentrations of species were set. The self-consistent plasmachemical radially symmetric model of a daytime sprite describes dynamics of 58 species: 25 neutral compounds including 9 in the excited state:

$O_2$, $N_2$, $O_3$, $CO$, $O$, $NO$, $CO_2$, $H$, $H_2O$, $N_2O$, $NO_2$, $NO_3$, $OH$, $N$, $HO_2$, $H_2$, $H_2O_2$, $O_2(a)$, $O_2(b)$, $N_2(A)$, $N_2(B)$,

$N_2(C)$, $N_2(a)$, $N(^2D)$, $O(^1D)$, $O(^1S)$,

10 negative ions and electrons:

$O_2^-$, $O_3^-$, $NO_3^-$, $CO_3^-$, $NO_2^-$, $NO^-$, $O_4^-$, $CO_4^-$, $OH^-$, $O^-$, $e$,

and 22 positive ions including 12 cluster ions:

$NO^+$, $H_2O^+$, $H_3O^+$, $O_2^+$, $O_4^+$, $H_3O^+(OH)$, $H_5O_2^+$, $N_2^+$, $N_4^+$, $N_3^+$, $O_2^+N_2$, $NO^+N_2$, $NO^+O_2$, $N^+$, $O^+$, $O_2^+(H_2O)$,

$H_7O_3^+$, $H_9O_4^+$, $(H_2O)NO^+$, $(H_2O)_2NO^+$, $(H_2O)_3NO^+$, $CO_2NO^+$.

The model describes 265 chemical reactions, the list of it is taken from the previous study with minor alterations (Evtushenko A.A., 2017). The photochemical block consists of 29 reactions adapted from the model (Winkler H., 2014). The model (Winkler H., 2014) which is based on the model (Clipperfield M.P., 1999), on the scheme proposed in (Lary D.J., 1991) and on the transport model (Winkler H., 2009). We decided not to add new species to the developed model for a nighttime

sprite, therefore, all species containing chlorine were not taken into account. Rates of photochemical reactions were calculated for analysed conditions by H. Winkler. In what follows numbers of photochemical reactions start with the letter P, numbers of all other reactions start with the letter R. The comprehensive list of plasmachemical and photochemical reactions is presented in



Supplementary Materials. The system of differential equations corresponding to the model includes the equations of chemical kinetics (Eq.1).

$$\frac{dN_i}{dt} = \sum_{k \in R_i} v_k \prod_{s \in J_k} N_s - \sum_{k \in L_i} v_k \prod_{s \in J_k} N_s \tag{1}$$


where $N_i$ is the concentration of a species number $i$, $v_k$ — rate of a reaction number $k$, $R_i$ is a set of reactions with species number $i$ at the right side, $L_i$ is a set of reactions with species number $i$ at the left side, $J_k$ is a set of indexes of species, which are reagents in the reaction number $k$. The first and second terms represent sources and sinks for a species number $i$ correspondingly.

The electric field in the sprite region is modeled self-consistently and is related to a lightning discharge in the troposphere according to the (Eq.2):

$$\frac{dE}{dt} + \frac{\sigma E}{\epsilon_0} = \frac{dE_{ext}}{dt}, \tag{2}$$

where $E_{ext}$ is electric field produced by the tropospheric uncompensated charge in the cloud, which occur during the flow of the electric current in the lightning chanel. $E$ — electric field at a specific point in the sprite region. $\sigma$ — conductivity, which

is defined by the (Eq.3,4).

$$\sigma = \frac{e^2 N_e}{m_e \nu_e(T_e)} \tag{3}$$

$$\nu_e(T_e) = 1.84 \cdot 10^9 \cdot \frac{N_m}{10^{17}} \cdot \left(\frac{T_e}{1000}\right)^{\frac{5}{6}}, \tag{4}$$

where $\nu_e(T_e)$ is the frequency of electron collisions, $N_m$ is the integral concentration of neutral species. $m_e$ and $e$ is electron mass and charge, $N_e$ is the electron concentration. We take into account only electron conductivity due to the fact that the ion

conductivity is negligibly small at an altitudes of interest. $T_e$ is the temperature of electrons, calculated using the BOLSIG+ free software package (*http://www.bolsig.laplace.univ-tlse.fr/*).

In order to calculate $E_{ext}$ we assume that the uncompensated charge in the cloud is distributed according to the normal law over a flat disk with a radius of 10 km, located at an altitude of 10 km above the Earth's surface. Earth is considered perfectly conductive and the electric field $E_{ext}$ is calculated in the dipole approximation. The charge on the disk is defined

as the integral of the current flowing through the lightning channel over time. The analysis of powerful positive discharges shows that the main increase in the ICM occurs in the first few ms (Stanley M., 2000). We use the model current dynamics which provides an increase in the ICM to a maximum value within 4 ms and smooth relaxation for 12 ms, corresponding to the processes of charge redistribution in the cloud. In this model the dependence of the ICM on the maximum current is linear. All





calculations were carried out for the city of Collioure in the south-west of France (42°31' N, 03°05' E), for 19:00 local time,
July 8.

Modeling is carried out in 3 steps:

1. In the framework of the proposed system of chemical reactions it is fundamentally impossible to achieve a completely
   equilibrium state, therefore, for 100 s the reaction system is calculated without disturbing the electric field and comes
   to a quasi-equilibrium state. Hereinafter, the time reference point is taken to be the start time of the current flow in the
lightning channel.

2. The system response to the perturbation of the electric field lasting 16 ms is calculated, which includes the process of
   current flow in the lightning channel with an increase in the uncompensated charge (4 ms) and complete relaxation of
   this charge (12 ms).

3. Dynamics of species within 1000 s after the sprite initiation are analysed. Consideration of relaxation processes at large
times can lead to incorrect conclusions, since relaxation processes can go on against the background of a change in con-
   centrations associated with the unsteadiness of the system. The relaxation of the system for relatively small perturbations
   of the chemical composition should be considered especially carefully. Diffusion is not taken into account in our model,
   since the estimates given in (Winkler H., 2014) show that at times up to 1000 s its influence is negligibly small.

As a start of the study a series of calculations was performed for discharges with a maximum ICM from 2000 to 4000 C·km
with a step of 25 C·km, which, taking into account the selected current profile, corresponds to the maximum current in the
lightning channel from 255 to 510 kA. The radius discretization step is 5 km, the altitude discretization step is 2 km. The current
values given above are high, but comparable with the maximum reliably measured — 280 kA (Goto Y., 1990). Indirect data
from the WWLLN show that events with currents of more than 300 kA occur quite often (Holzworth R.H., 2019). The analysis
of the simulation results lead to the conclusion that in the selected range of ICM there are 2 fundamentally different scenarios
of system behavior. At relatively small ICM, the electric field reaches 100–110 Td, which is lower than the breakdown field, but
nevertheless sufficient for luminescence and perturbation of the chemical balance. An important feature of this regime is the
absence of an avalanche of electrons due to their fast attachment to molecular oxygen. This regime represents the diffuse glow
when the field is not strong enough for the development of the streamer structure. Simulation of this behavior of the system was
conducted for the ICM of 2750 C·km, corresponding to a maximum field of 108 Td. The second simulated scenario (for the
ICM of 3750 C·km and the maximum normalised field of 128 Td) includes strong perturbations of the electron concentration,
i.e. development of avalanche ionization. The considered ICM values are typical of measured characteristics of daytime sprites
(Stanley M., 2000). The maximum value of the normalized electric field is observed at an altitude of 62 km, therefore, the main
study of the perturbation of the chemical composition will be carried out for this altitude. In what follows the calculation results
for points on the discharge axis correspond to the altitude value of 62 km unless the opposite is indicated. The simulation for
ICM of 2750 C·km and 3750 C·km was carried out with the radius discretization step of 1 km and the altitude discretization
step of 1 km.



## 3   Simulation for ICM = 2750 C·km

The main part of negative charge in the daytime at the altitudes of interest is located on the electrons and $CO_3^-$, $O_2^-$. In the quasi-equilibrium state the electron concentration remains approximately constant due to photochemical detachment from $O_2^-$


$$O_2^- \rightarrow e + O_2 \tag{P18}$$

and the interaction of $O_2^-$ with $O$ and $O_2(a)$

$$O_2^- + O \rightarrow O_3 + e \tag{R149}$$

$$O_2^- + O_2(a) \rightarrow O_2 + O_2 + e \tag{R144}$$

The sink is provided by attachment to molecular oxygen and interaction with ozone, which leads to production of $O_2^-$.

$$e + O_2 + O \rightarrow O_2^- + O \tag{R166}$$

$$e + O_3 \rightarrow O_2^- + O \tag{R169}$$

Thus, a quasi-equilibrium state of electron and $O_2^-$ is maintained with an electron concentration of 95 cm$^{-3}$ and $O_2^-$ concentration of 62 cm$^{-3}$ at an altitude of 62 km. The concentration of $CO_3^-$ is maintained due to the main reaction of $O_3^-$ and $CO_2$.

$$O_3^- + CO_2 \rightarrow CO_3^- + O_2 \tag{R150}$$

The sink is related to the formation of $O_2^-$ by interaction with $O$

$$CO_3^- + O \rightarrow O_2^- + CO_2 \tag{R120}$$

and photochemical decomposition with the formation of $O^-$

$$CO_3^- \rightarrow O^- + CO_2 \tag{P22}$$

Concentrations of all other negative ions in the quasi-equilibrium state are small. Almost all the positive charge is on the heaviest of the considered complex ions — $H_9O_4^+$ which has a concentration of 175 cm$^{-3}$. Described processes maintaining

quasistationary state (corresponding to the first step of modeling) are independent on the electric field applied before a sprite initiation (on the second step of modeling).





At the initial stage of the discharge the electric field increases, and the main sources of electrons are reactions of the interaction of electrons with molecular nitrogen and oxygen, the rates of which depend on the normalized electric field.

$$e + N_2 \rightarrow N_2^+ + e + e \qquad \text{(R17)}$$


$$e + O_2 \rightarrow O_2^+ + e + e \qquad \text{(R15)}$$

The interaction of electrons and nitrogen is about 2 times more effective than electron interaction with oxygen, due to a higher concentration of nitrogen and a lower ionisation rate in the electric field during the development of a sprite. The accumulation of electron concentration does not occur due to the disappearance of free electrons due to the reaction of $O_2$ with

electrons in the electric field with the formation of $O^-$.

$$e + O_2 \rightarrow O^- + O \qquad \text{(R19)}$$

As a result, the electron concentration drops from 95 cm$^{-3}$ almost to zero (Fig.1). The indicated effective channel for dissociative attachment does not lead to the formation of significant concentration of $O^-$, since it rapidly interacts with molecular oxygen, producing $O_2^-$.

$O^- + O_2 \rightarrow O_2^- + O \qquad \text{(R136)}$

Thus, by 10 ms, almost all electrons are attached to oxygen, and at the same time the electric field is significantly decreased. Further, in the absence of the strong electric field detachment of the charge from $O_2^-$ begins in reactions with $O$, $O_2(a)$ and in photochemical detachment.

$$O_2^- + O \rightarrow O_2 + O^- \qquad \text{(R148)}$$


$$O_2^- + O \rightarrow O_3 + e \qquad \text{(R149)}$$

$$O_2^- + O_2(a) \rightarrow O_2 + O_2 + e \qquad \text{(R144)}$$

$O_2^- \rightarrow e + O_2 \qquad \text{(P18)}$

The electron and $O_2^-$ concentrations return to unperturbed values by 0.5 s. The interaction of $O_2^-$ with $O_3$ leads to the production of $O_3^-$, which is rapidly transformed into $CO_3^-$ by interaction with $CO_2$.

$$O_2^- + O_3 \rightarrow O_3^- + O_2 \qquad \text{(R147)}$$



$O_3^- + CO_2 \rightarrow CO_3^- + O_2$ (R150)

The perturbation of concentration of $CO_3^-$ is 10% and completely relaxes to the unperturbed state in 200 s (Fig.1).

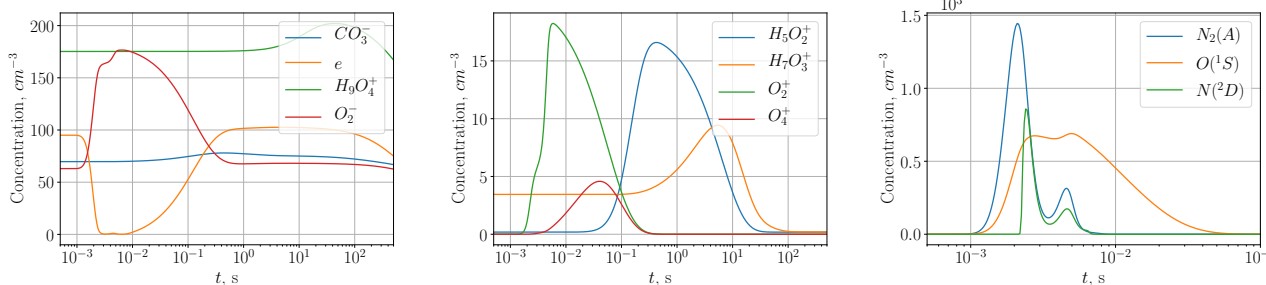

**Figure 1.** Dynamics of the concentration of the main ions, electrons and excited states of nitrogen and oxygen at an altitude of 62 km.

Ionization of nitrogen and oxygen during the increase of electric field strength is the main source of positive ions during discharge .

$e + N_2 \rightarrow N_2^+ + e + e$ (R17)


$e + O_2 \rightarrow O_2^+ + e + e$ (R15)

$N_2^+$ does not accumulate due to the rapid interaction with $O_2$.

$N_2^+ + O_2 \rightarrow O_2^+ + N_2$ (R65)

By 6 ms the $O_2^+$ concentration reaches a maximum value of 18 cm$^{-3}$ and by 0.2 s it relaxes to unperturbed values. Sink of
$O_2^+$ concentration is related to the formation of $O_2^+ N_2$ and $O_4^+$.

$O_2^+ + N_2 + N_2 \rightarrow O_2^+ N_2 + N_2$ (R47)

$O_2^+ + O_2 + O_2 \rightarrow O_4^+ + O_2$ (R52)

Both ions are transitional in the process of production of positive cluster ions, in which the concentration of $O_2^+ N_2$ is
almost constant, and the concentration of $O_4^+$ has a maximum value of 4.5 cm$^{-3}$. The concentration of $H_3O^+$ reaches the
value of 1.6 cm$^{-3}$, $H_5O_2^+$ — 16.5 cm$^{-3}$ to 0.3 s. The concentration of $H_7O_3^+$ reaches the value of 9.5 cm$^{-3}$ at 9 s. The
concentration of $H_9O_4^+$ is perturbed from 175 to 202 cm$^{-3}$ at 35 s , which is followed by relaxation to the unperturbed value



for 200 s. The described sequence of reactions of cluster ions production was considered in detail in (Evtushenko A.A. (2014), Evtushenko A.A. (2017)) and proceeds similarly under the conditions considered here.

The increase of the normalized electric field strength leads not only to an acceleration of the ionization of molecular nitrogen and oxygen, but also to the production of a large number of molecules and atoms in an excited state. When electrons interact with molecular nitrogen, $N_2(A)$, $N_2(a)$, $N_2(B)$, and $N_2(C)$ molecules are formed.

$$e + N_2 \rightarrow e + N_2(A) \tag{R5}$$

$$e + N_2 \rightarrow e + N_2(a) \tag{R7}$$

$$e + N_2 \rightarrow e + N_2(B) \tag{R6}$$

$$e + N_2 \rightarrow e + N_2(C) \tag{R8}$$

There is no significant perturbation of the concentration of $N_2(C)$ due to the fast radiation processes in the second positive band of molecular nitrogen and production of $N_2(B)$, the concentration of which is only slightly perturbed due to radiation processes in the first positive band of molecular nitrogen.

$$N_2(C) \rightarrow N_2(B) + h\nu(2PN_2) \tag{R236}$$

$$N_2(B) \rightarrow N_2(A) + h\nu(1PN_2) \tag{R233}$$

    Some accumulation is observed only for $N_2(A)$: its concentration reaches $1400\ \mathrm{cm^{-3}}$ by 2 ms, which is followed by a quick nonradiative relaxation to an unperturbed state by 7 ms. The ratio of density of optical radiation in the first and second positive bands of nitrogen reaches the value of 5 in the maximums of radiation. The interaction of electrons with molecular nitrogen in the electric field leads to the formation of $N(^2D)$ with a concentration of $800\ \mathrm{cm^{-3}}$ 2.5 ms after the start of the sprite.

Dynamics of concentration of all the described species are presented at Fig.1.

    The relaxation of concentration of $N(^2D)$ takes place simultaneously with the relaxation of concentration of $N_2(A)$, because $N_2(A)$ is the main source for $N(^2D)$ and atomic nitrogen.

$$N_2(A) + O_2 \rightarrow NO + N(^2D) \tag{R219}$$

$$N(^2D) + N_2 \rightarrow N + N_2 \tag{R242}$$



$$N(^2D) + O \rightarrow N + O(^1D) \tag{R243}$$

By the end of the discharge, the concentration of $N$ is increased by a factor of 3, the relaxation time to the unperturbed state is about 300 seconds. On the plots of the concentration dynamics of $N_2(A)$ and $N(^2D)$ 2 peaks are observed: at 2.5 ms and 4.5 ms, which is explained by the following. Up to 4 ms, the field strength increases, the concentration of $N_2(A)$ and $N(^2D)$ increases to 2.5 ms. A decrease in the electron concentration due to attachment to oxygen leads to a decrease in concentrations of $N_2(A)$ and $N(^2D)$ by almost an order of magnitude from the maximum value, while the field strength is close to the maximum value. A slight decrease in the field strength at the relaxation stage rapidly slows down the attachment of electrons to molecular oxygen, which leads to an increase in the efficiency of production of $N_2(A)$ and $N(^2D)$ and the formation of a second peak by 4.5 ms. Then, the electric field decreases, and the perturbation of the concentration of excited nitrogen rapidly relaxes. The interaction of electrons with molecular oxygen in an electric field leads to the production of a significant amount of $O(^1S)$ and $O(^1D)$.

$$e + O_2 \rightarrow e + O + O(^1S) \tag{R14}$$

$$e + O_2 \rightarrow e + O + O(^1D) \tag{R13}$$

Despite the fact that $O(^1S)$ transformes into $O(^1D)$, this does not lead to the maintenance of a perturbation of concentration of the $O(^1D)$ after the field relaxation, due to its rapid nonradiative quenching on molecular nitrogen and oxygen.

$$O(^1D) + N_2 \rightarrow O + N_2 \tag{R247}$$

$$O(^1D) + O_2 \rightarrow O + O_2 \tag{R248}$$

A noticeable change in the electron concentration occurs at altitudes from 58 to 64 km, where the electron concentration decreases sharply due to the chain of reactions leading to the formation of $O_2^-$. The maximum perturbation of the $O_2^-$ concentration occurs at an altitude of 63.5 km and reaches $265$ cm$^{-3}$. After 1 s, the concentration of electrons and $O_2^-$ in the entire range of heights returns to unperturbed values (Fig.2). A significant perturbation of $O_2^+$ concentration occurs in the range of altitudes from 59.5 km to 63.5 km. By 0.1 s, the concentration of $O_2^+$ completely relaxes mainly due to the formation of cluster ions. The perturbation of $H_5O_2^+$ behaves in a similar way, but relaxation lasts up to 30 s (Fig.3). The concentration of $H_9O_4^+$ is smoothly perturbed, reaches a maximum by 70 s and then relaxes to the initial value.

Of the neutral compounds, atomic nitrogen is most significantly and long-term perturbed at altitudes from 59 to 63.5 km, with a maximum of $3.4 \cdot 10^4$ cm$^{-3}$ per 62 km. The relaxation of N concentration goes smoothly and takes about 300 s. The




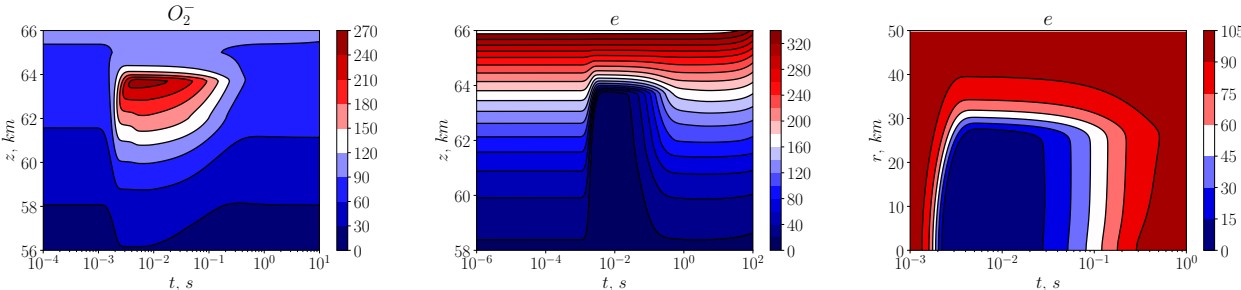

**Figure 2.** The altitude dependence of $O_2^-$ concentration, the altitude and radial dependence of electron concentration.

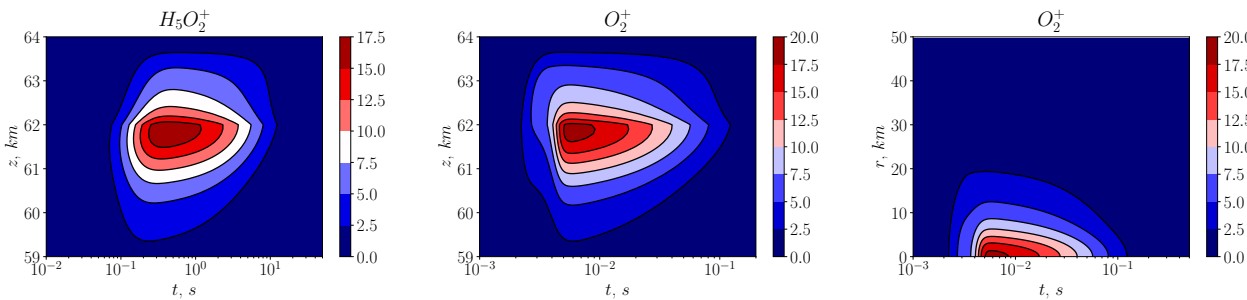

**Figure 3.** The altitude dependence of $H_5O_2^+$ concentration, the altitude and radial dependence of $O_2^+$ concentration.

perturbation of concentrations of the excited speies $N_2(A)$, $O(^1S)$, $O(^1D)$, $N(^2D)$ occurs within the altitude range from 59 km to 64 km (Fig.4). The concentration of $N_2(A)$ reaches 1850 cm$^{-3}$ at the altitude value 63 km, but relaxes very quickly with the beginning of the field decrease. The lifetime of $N(^2D)$ is even shorter, its concentration is significant only in time range shorter than 1 ms. A similar dynamics is shown by $O(^1D)$, although the perturbation is much smaller in amplitude. The perturbation of concentration of $O(^1S)$ reaches the value of 1200 cm$^{-3}$ and relaxes in 30 ms at an altitude of 63.5 km (Fig.4).

The growth of the normalized electric field goes smoothly and synchronously with the accumulation of an uncompensated charge in the cloud: by 4 ms after the start of the sprite, a maximum value of 108 Td is reached, which is below the critical field of 128 Td, but it is sufficient for plasmachemical reactions (Fig.5). A smooth decrease in the field from the maximum value lasts 12 ms. The maximum normalized field of about 108 Td is reached in the altitude range from 61 to 62.5 km for 1 ms, which leads to a perturbation of the chemical composition. The discussed lightning discharge parameters correspond to the development of a very weak sprite or halo. The electron temperature gradually increases to $2.9 \cdot 10^4$ K along with an increase in the electric field, while the temperature of the neutral species does not change. Along with an increase in the electric field, conductivity begins to decrease due to an increase in electron temperature and collision frequency. At altitudes of 63 km and lower, a sharp drop in the electron concentration occurs, which leads to an additional decrease in conductivity by 2–3 orders of magnitude. At an altitude of 64 km, a drop in electron concentration does not occur; the change in conductivity is much smaller. The weakening of the field leads to a drop in the electron temperature and a rapid restoration of conductivity, and the





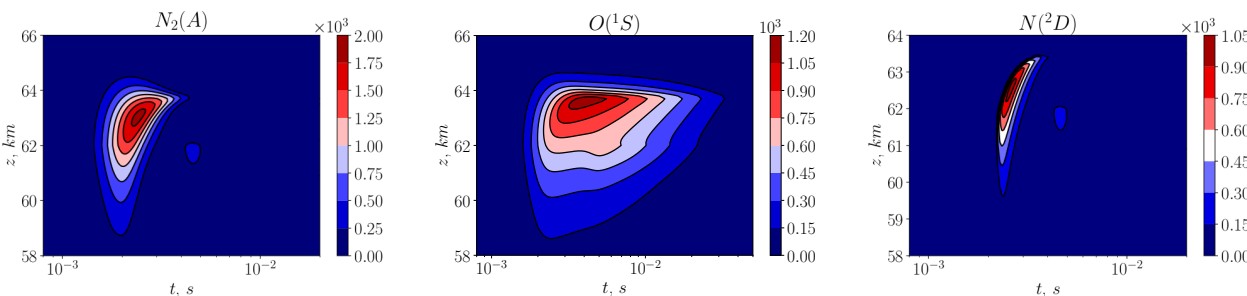

**Figure 4.** The altitude dependence of $N_2(A)$, $O(^1S)$, $N(^2D)$ concentration.

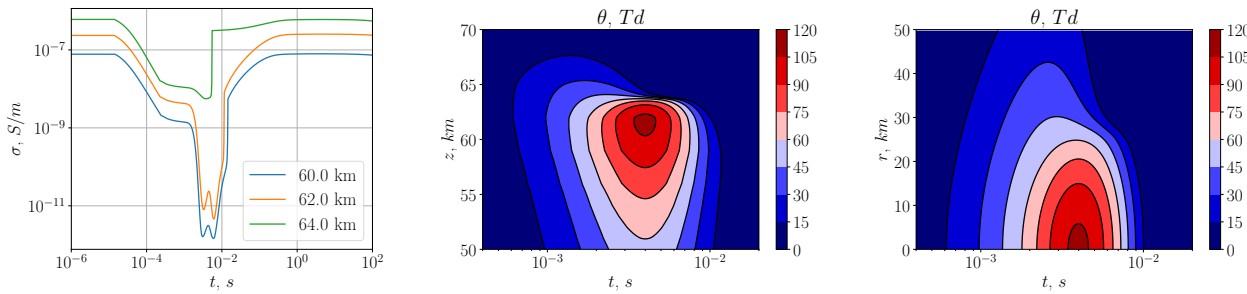

**Figure 5.** Conductivity dynamics on the different altitudes, the altitude and radial dependence of the normalised electric field.

relaxation of the electron concentration after 1 s completely restores the conductivity (Fig.5). The field gradually decreases in the radial direction, taking values above 105 Td at a distance of up to 5 km from the discharge, and above 75 Td at a distance of up to 20 km. Concentrations of species are perturbed in the region with a diameter of about 40–50 km and altitude range from 59 to 64 km (Fig.2, 3, 5). The most intense radiation from the discharge is due to the first and second bands of molecular nitrogen and occurs at altitudes from 59 km to 64 km, having maximum slightly below 63 km. Time dependences of radiation in the first and second bands have a similar shape, while the maximal intensity in the first positive band is up to 6 times greater than in the second (Fig.6).

## 4   Simulation for ICM = 3750 C·km

The maximum current of the lightning channel at the same time parameters for the ICM of 3750 C·km is proportionally higher than for the ICM of 2750 C·km. The normalized field at an altitude of 62 km reaches 128 Td, which creates the conditions for the formation of an electron avalanche. The dependence of the maximum normalized field in the sprite region on the maximum ICM is not linear due to the self-consistency of the simulation. The normalized electric field is maximal at altitudes of 62–64 km; that is why dynamics of species concentration will be discussed in detail for an altitude of 62 km, similarly to the previous simulation case.





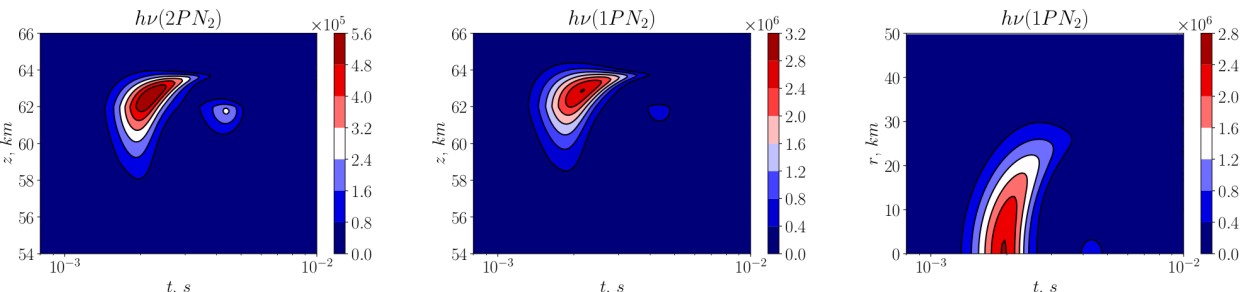

**Figure 6.** Dynamics of the altitude and radial distribution of emission rate in the first and the second positive nitrogen bands.

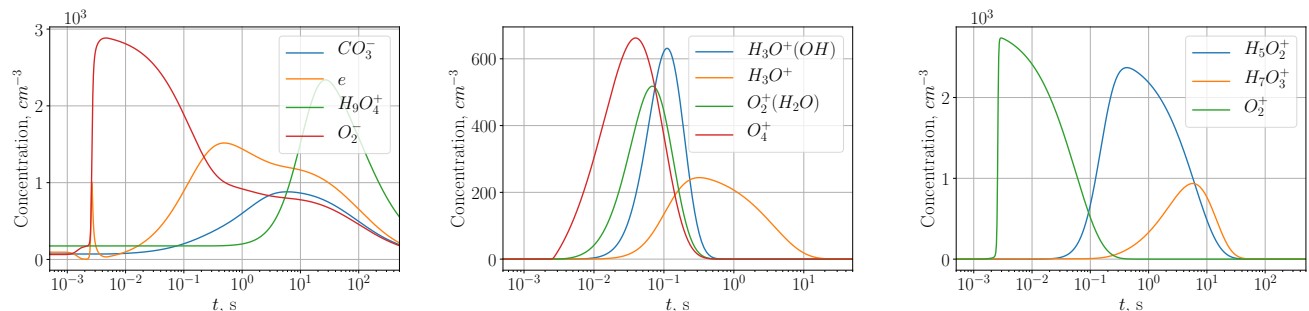

**Figure 7.** Concentration dynamics of ions and electrons at the altitude of 62 km.

At the beginning of the discharge, field increase leads to the rapid drop of electron concentration due to the attachment of electrons to molecular oxygen with the formation of $O_2^-$ similar to the case of ICM of 2750 C·km. A further increase of the field to breakdown values leads to the formation of an avalanche of electrons and a sharp increase of the electron concentration to 1400 cm$^{-3}$, and the concentration of $O_2^-$ to 1500 cm$^{-3}$ (Fig.7). As soon as the field becomes lower than the breakdown,

electron attachment rate exceeds free electron production rate, which leads to a second wave of a decrease of the electron concentration, which drops to almost zero by 5 ms. At this moment, almost all the negative charge is accumulated on $O_2^-$, a maximum concentration of which is 2800 cm$^{-3}$ for 4 ms. The concentration of $O^-$ sharply increases when electric fields becomes critical, and reaches the maximum value of 100 cm$^{-3}$ for a time interval of about 0.5 ms. $O^-$ ion is an intermediate in the chain between electrons and $O_2^-$, and the perturbation of its concentration quickly relaxes.

$$O^- + O_2 \rightarrow O_2^- + O \tag{R136}$$

$O_2^+$ appears during the sprite and reaches a concentration of 2800 cm$^{-3}$ by 2 ms, accounting for almost the entire positive charge. $O_2^+$ is the ion from which the chain of reactions begins, leading to the formation of a family of cluster ions. At the relaxation stage, concentrations of the species $O_4^+$, $O_2^+(H_2O)$, $H_3O^+(OH)$ increase to 500–600 cm$^{-3}$ to 0.05–0.1 s, with complete relaxation to 0.5 s (Fig.7). This is followed by the formation of a family of cluster ions of the form $H^+(H_2O)_n$. The

duration of the perturbation process is longer for heavier ions: $H_3O^+$ with a maximum of about 200 cm$^{-3}$ to 0.5 s, $H_5O_2^+$





with a maximum of 2200 cm$^{-3}$ at the same times, $H_7O_3^+$ with a maximum of 950 cm$^{-3}$ to 7 s, and long-lived $H_9O_4^+$ ion with a maximum of 2300 cm$^{-3}$ to 30 seconds after the start of the sprite. The relaxation of the negative charge after the sprite collected on $O_2^-$ occurs together with the detachment of electrons and the formation of $CO_3^-$. The maximum electron concentration of 1250 cm$^{-3}$ is reached by 0.5 s, and the concentration of $CO_3^-$ reaches 900 cm$^{-3}$ by 8 seconds.

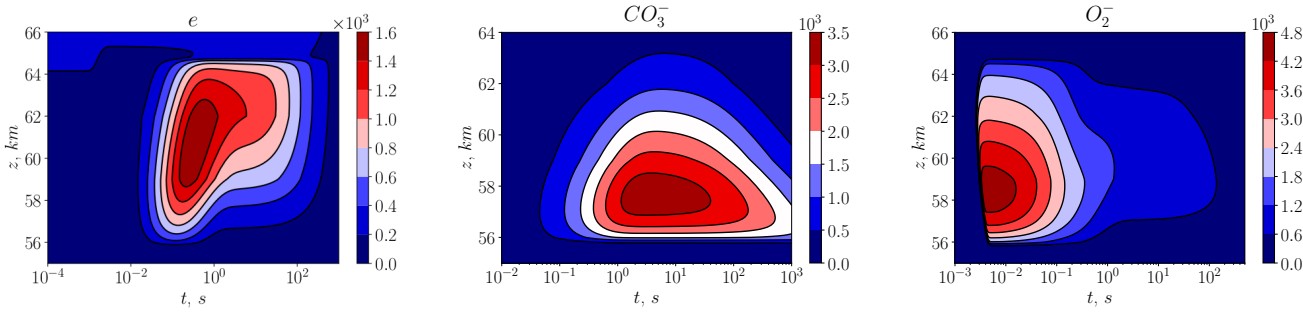

**Figure 8.** The altitude dependence of electrons, $CO_3^-$ - and $O_2^-$ concentration.

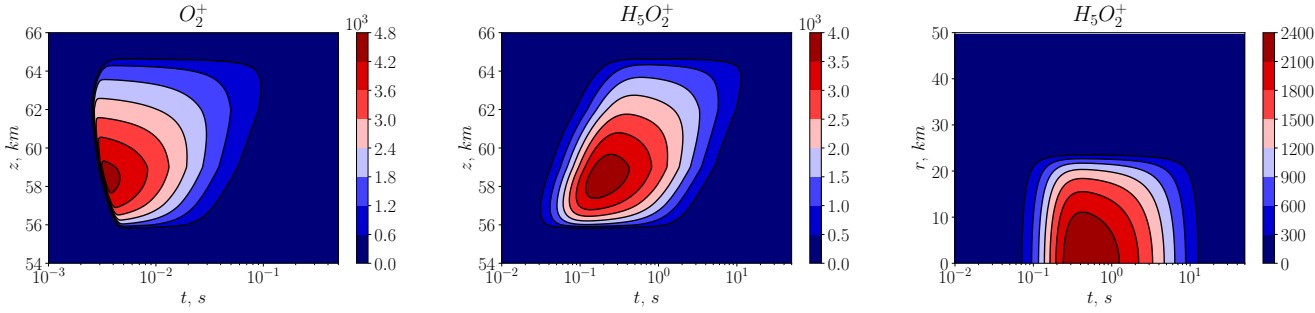

**Figure 9.** The altitude dependence of $O_2^+$ and $H_5O_2^-$ concentration, the altitude and radial dependence of $H_5O_2^+$ concentration.

In comparison with the previous modeling conditions, the sprite area has a larger size both in radius and in height. The electron concentration is perturbed at altitudes from 56 to 64 km and dynamics are similar to the case of altitude of 62 km. Maximum concentrations of electrons more than 1400 cm$^{-3}$ are reached at altitudes from 58.5 to 62.5 km (Fig.8). The characteristic time of perturbation and relaxation of the electron concentration increases with altitude. Concentration of $O_2^-$ reaches the value of more than 4200 cm$^{-3}$ at altitudes of 57.5–59.5 km, after about 1 s its concentration falls below 1000 cm$^{-3}$, 330    after 100 s at all altitudes the perturbation completely relaxes. The maximum perturbation of concentration of $CO_3^-$ is shifted downward relative to electron maximum to the altitude of 57–59 km. The concentration of $CO_3^-$ is more than 3000 cm$^{-3}$ for several tens of seconds. Complete relaxation takes 1000 s in the upper part and more than 1000 seconds in the lower part (Fig.8). During the discharge, a positive charge at altitude of 56–64 km accumulates on $O_2^+$. Maximum value of $O_2^+$ concentration is 4800 cm$^{-3}$ and occurres at the altitude of 58 km. The sequence of reactions producing cluster ions decreases the 335    $O_2^+$ concentration almost to zero. The concentration of $H_5O_2^+$ exceeds the value of 3500 cm$^{-3}$ in the altitude range 57.5–59.5





km. The rate of production of cluster ions decreases with altitude, as well as the relaxation rate (Fig.9). At around 1000 s, the perturbation of the ionic composition caused by the sprite completely relaxes.

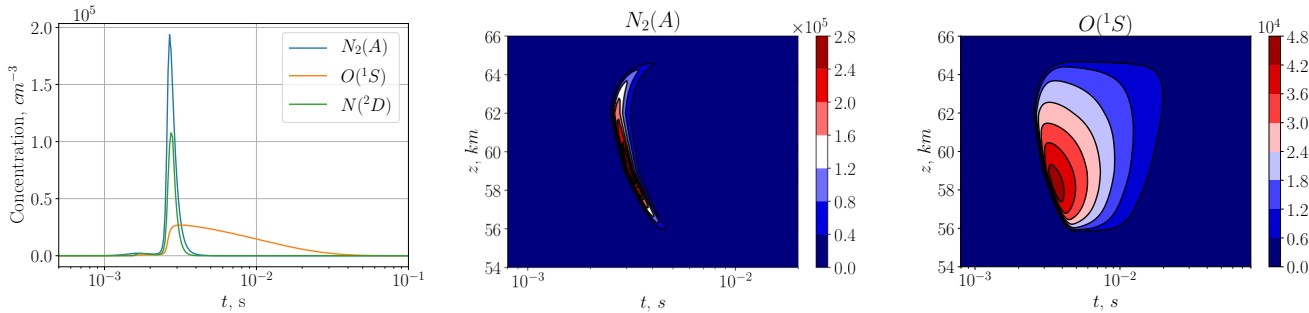

**Figure 10.** Dynamics and the altitude dependence of the excited molecules $N_2(A)$, $O(^1S)$, $N(^2D)$.

Concentrations $N_2(A)$ and $N(^2D)$ are most perturbed relative to other excited states, up to about $2 \cdot 10^5$ cm$^{-3}$ and $10^5$ cm$^{-3}$, respectively, and with a short lifetime of about 2 ms. The concentration of $O(^1D)$ has a similar dynamics, but a maximal

variation of concentration is significantly lower than that for $N_2(A)$. The perturbation of the $O(^1S)$ concentration lasts up to 30 ms and reaches $2 \cdot 10^4$ cm$^{-3}$ (Fig.10).

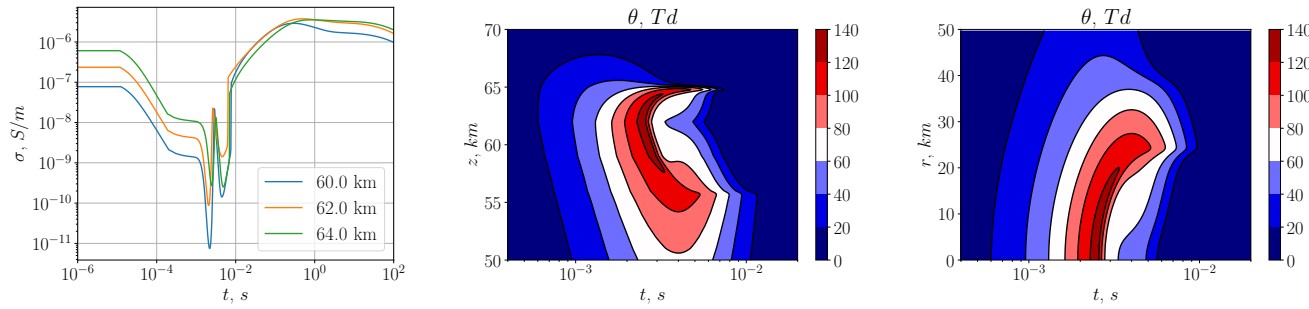

**Figure 11.** Dynamics of the conductivity and distribution of the normalised electric field.

Dynamics of the normalized electric field look alike in all the range of altitudes, but the moment of the field maximum and the duration of relaxation are dependent on the altitude (Fig.11). In the region of 58–64 km, where a field of more than 120 C·km is observed and conditions are created for the formation of an electron avalanche, a sharp increase in conductivity by

about 2–3 orders of magnitude occurs, which leads to a rapid displacement of the field (Fig.11). After this, the conductivity decreases with the attachment of electrons, and then with a decrease in the field, an increase in the conductivity at these altitudes follows. Conditions for the formation of an electron avalanche are created at a distance of up to 20 km from the discharge axis, and at a distance of 10 km more the electric field reaches 100–110 Td, i.e. the conditions described in the first part of the article take place. A field at the distance of 20 to 30 km from the discharge axis is perturbed about 2 times longer than on the axis,

because there is no displacement due to a sharp increase in conductivity.





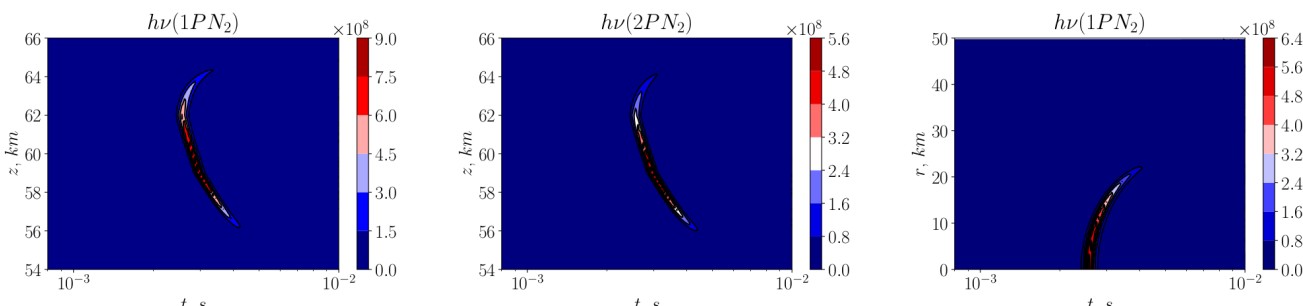

**Figure 12.** Dynamics of the altitude and radial distribution of emission rate in the first and the second positive nitrogen bands.

An increase in the field in the discharge with ICM of 3750 C·km leads to a significant increase in the intensity of optical emissions. As in the previous modeling case, the main glow occurs in the first and second positive band of nitrogen. Intense radiation is observed in the region of strong field. The maximum optical emission rate in the first and second nitrogen bands differ by about 2 times. The radiation power of the sprite in the radial direction is significant at a distance from the discharge
axis of less than 20 km, then the emission rate drops sharply with the normalized field.

## 5 Conclusions

The paper considers sprite modeling in daytime conditions using a self-consistent axially symmetric plasmachemical model. It is shown that, in order to initiate a high-altitude discharge, it is necessary to create in the cloud the ICM of several thousand C·km, which is in a good agreement with the measurements reported in (Stanley M., 2000). Conditions for initiating a daytime
sprite are created at altitudes 20 km lower than for night sprites. The diffuse region of a daytime sprite occupies a range of altitudes of 56–64 km and is considered in detail in the article. The shape of the daytime sprite is different from that of the night sprite: the horizontal size is comparable, while the height is smaller. The streamer part of a daytime sprite is located lower and cannot be considered in the approximation used for the electric field due to the extremely strong spatial inhomogeneity of the field. During modeling, it was found that 2 scenarios of the discharge development are possible, depending on the strength of
the normalized electric field at the altitude of 50–70 km. For the ICM of 2750 C·km, the electric field value remains below the threshold, i.e. sufficient for a slight perturbation of the chemical balance and radiation in the first and second band of molecular nitrogen, but insufficient for the creation of avalanche ionization. Most likely, for the specified parameters, the streamer part is not formed and this discharge can be characterized as an incompletely formed sprite or halo. An increase in the lightning power in the troposphere and an increase in the ICM to 3750 C·km leads to the implementation of the second scenario —
avalanche ionization with much stronger chemical disturbances and a longer stage of their relaxation. In the strong field region the formation of avalanche ionization results in a quick displacement of the field due to an increase in conductivity, leading to the necessity of a self-consistent consideration of the problem.



Authors thank Holger Winkler for the provided data on photochemical reactions and useful discussions during the preparation of the article.

This research has been supported by the Ministry of science and higher education of the Russian Federation, agreement No. 075-15-2019-1892.

*Code availability.* Simulation code is available at https://bitbucket.org/spritelab269/daytimespritemodeler2019/src/master/.

*Author contributions.* AE designed the study, FK performed the numerical simulation, AE and ES conducted the data analysis and wrote the paper.

*Competing interests.* The authors declare that they have no conflict of interest.



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
