# Peer review of "A plasmachemical axially symmetric self-consistent model of daytime sprite"

_Atmospheric Chemistry and Physics, 2019_

## Referee Comment (RC1) · Anonymous Referee #2 · 5 May 2020

This paper in on sprite modeling in daytime conditions using a self-consistent axially symmetric plasmachemical model. The authors have first given a good background and observational evidence of nighttime sprites which high frequency of their occurrence. They have then explained the difficulty with direct observations of the daytime sprites and their less frequent occurrence because of requirement of large impulse charge moment (ICM). They have also given the references of indirect evidences of daytime sprites. The authors have done the simulation for two values of ICM of 2750 C_km and 3750 C km with the radius discretization step of 1 km and the altitude discretization step of 1 km which is a reasonable consideration at a reduced height by 20 km as compared to nighttime sprites which I consider technically valid hence results and conclusions are sound

[Figure]

I just have two corrections and additions to be made:

1. Lines : "Similar pattern of the magnetic field evolution was observed in daytime and is presumably related to sprites (Kumar S., 2008)". I suggest authors to read this article and rewrite again appropriately as there were daytime VLF perturbations observed from Electric field measurements of VLF subionospheric waves from navigational transmitters not the magnetic field.

2. Lines 132-133: "Indirect data from the WWLLN show that events with currents of more than 300 kA occur quite often (Holzworth R.H., 2019)". I suggest here to cite at least these two following relevant studies have used GLD360, WWLLN and NLDN data .

• Salut, M. M., M. Abdullah, K. L. Graf, M. B. Cohen, B. R. T. Cotts, and S. Kumar (2012), Long recovery VLF perturbations associated with lightning discharges, J. Geophys. Res., 117, A08311, doi:10.1029/2012JA017567. • Salut, M. M., M. B. Cohen, M. A. M. Ali, K. L. Graf, B. R. T. Cotts, and S. Kumar (2013), On the relationship between lightning peak current and Early VLF perturbations, J. Geophys. Res. Space Physics, 118, doi:10.1002/2013JA019087.

I am then happy to recommend the publications of this paper.

Please also note the supplement to this comment:
https://www.atmos-chem-phys-discuss.net/acp-2019-1196/acp-2019-1196-RC1-supplement.pdf

---

## Author Comment (AC1) · 3 Jun 2020

The authors are grateful to Anonymous Referee #2, whose comments helped to improve the manuscript. The following changes are made:

1. The phrase which refers to (Kumar S., 2008) is corrected: "magnetic field evolution" is changed to "VLF perturbations".

2. The two proposed citations are added:

Properties of VLF perturbations in association with lightning discharges are discussed in detail in Salut M.M.(2012). The relation between the scattering pattern, recovery duration, and occurrence rate of VLF events with the peak current magnitude and polarity of the causative lightning discharges is studied in Salut M.M.(2013).

---

## Referee Comment (RC2) · Anonymous Referee #1 · 1 Jul 2020

**1   Overview**

This paper by A. Evtushenko, F. Kuterin and E. Svechnikova focuses on the chemistry of daytime sprites. Little is known about daytime sprites, and their existence is only hypothesized from a handful of radio observations. Unfortunately, in its present form, the work by Evtushenko et al. contributes little to our further understanding of these phenomena and contains several issues that prevent me for recommending its publication in ACP.

The paper presents modeling results but it does not connect them to any existing observation. It also does not indicate a realistic path where the model can contribute to explain a future observation and thus elucidate the physical mechanism behind the radio disturbances above thunderstorms. There is no discussion on whether the modeled changes in conductivity would explain explain the observed radio disturbances. Such is the kind of discussion that would contribute to our understanding of these phenomena.

I should also comment on the use of the term "sprite" in the title and through the manuscript. Sprites are electrical discharges above thunderstorms composed by thin ionized plasma filaments. These filaments, called streamers, propagate because the small radius of curvature at their tips strongly enhances the electric field and induce strong impact ionization. The electrostatic model proposed in this manuscript (see equation (2)) disregards this curvature and therefore it is misleading to talk about sprites.

The discussion presented here is more appropriate for halos: diffuse, extended discharges where the curvature of the charge region plays a minor role. The authors call this "diffuse mode of propagation" and it is true that we do not know for sure if the handful of radio signals mentioned above are caused by sprites, halos or something different altogether. However, the electric fields selected by the authors are probably too high for halos: any small inhomogeneity in the atmospheric conductivity would locally drive the electric field above the effective ionization threshold and trigger the propagation of streamers.

Since a sketch of the atmospheric chemistry involved in a daytime sprite/halo was already presented by Winkler and Notholt (2014), the claim to novelty in the manuscript by Evtushenko et al. is the self-consistent computation of electric field. However, the validity of this computation is not clearly established in the paper for the reasons mentioned above.

**2 Additional issues**

Some other, more easily fixable but still relevant problems that I found in the manuscript are:

- line 47: Please include references for the conductivity of the daytime ionosphere.

- line 70: "reactions are left beyond the framework of consideration." What reactions? Why? How do we know if they are relevant?

- line 85: "all species containing chlorine were not taken into account": Should they? Again, how do we know?

- line 90: This expression does not include stoichiometric coefficients when they are not equal to one. For example the right side of reaction (R17) contains 2 electrons.

- equation (4): This equation lacks units and a reference.

- line 107: The distance between the thundercloud and the discharge here is shorter than for normal sprites. Is it still acceptable to disregard the charge structure in the cloud? What about the induction field? Can it also be neglected here?

- line 117: Please specify what is meant by "quasi-equilibirum state". Is the electron density stationary after relaxation? You should compare the electrical properties of air at that point with atmospheric measurements to ensure that the computations are realistic. It would also be interesting to see the evolution of the unperturbed system for 1000 s to see if parts of the perturbation attributed to the discharge are present also in a freely evolving system.

- line 138: It is not true that streamers do not develop below the conventional breakdown electric field: they are easily triggered by inhomogeneities and can propagate in fields as low as ~20 Td.

**3 Detailed review criteria**

**3.1 Scientific significance**

I think that the paper is not a substantial contribution to scientific progress. It does not contain any new concepts, ideas, methods, or data. The methods in the paper have been applied before in other contexts and their validity in the present context is unclear and not carefully discussed.

**3.2 Scientific quality**

It is not clear if the proposed methods are valid. The model is valid for halos but the electric field is too high for a halo-only discharge. We do not know if the radio signatures that motivate this paper are due to sprites, halos or something different. The authors do not discuss this.

**3.3 Presentation quality**

The presentation quality is generally good although some aspects can be improved. There are a few awkward sentences and the legibility of the figure labels could be improved.